# Intraoperative Laparoscopic Hyperspectral Imaging during Esophagectomy—A Pilot Study Evaluating Esophagogastric Perfusion at the Anastomotic Sites

**DOI:** 10.3390/bioengineering11010069

**Published:** 2024-01-09

**Authors:** Annalena Ilgen, Hannes Köhler, Annekatrin Pfahl, Sigmar Stelzner, Matthias Mehdorn, Boris Jansen-Winkeln, Ines Gockel, Yusef Moulla

**Affiliations:** 1Department of Visceral, Transplant, Thoracic and Vascular Surgery, University Hospital of Leipzig, Liebigstr. 20, 04103 Leipzig, Germany; annalena.ilgen@gmx.de (A.I.); sigmar.stelzner@medizin.uni-leipzig.de (S.S.); ines_gockel@hotmail.com (I.G.); 2Innovation Center Computer Assisted Surgery (ICCAS), Faculty of Medicine, Leipzig University, Semmelweisstr. 14, 04103 Leipzig, Germany; hannes.koehler@medizin.uni-leipzig.de (H.K.); annekatrin.pfahl@medizin.uni-leipzig.de (A.P.); 3Department of General, Visceral, Thoracic and Vascular Surgery, Klinikum St. Georg, Delitzscher Str. 141, 04129 Leipzig, Germany; boris.jansen-winkeln@sanktgeorg.de

**Keywords:** hyperspectral imaging, minimally invasive/robotic surgery, clinical evaluation study, gastrointestinal surgery, intraoperative imaging, esophagectomy

## Abstract

Hyperspectral imaging (HSI) is a non-invasive and contactless technique that enables the real-time acquisition of comprehensive information on tissue within the surgical field. In this pilot study, we investigated whether a new HSI system for minimally-invasive surgery, TIVITA^®^ Mini (HSI-MIS), provides reliable insights into tissue perfusion of the proximal and distal esophagogastric anastomotic sites during 21 laparoscopic/thoracoscopic or robotic Ivor Lewis esophagectomies of patients with cancer to minimize the risk of dreaded anastomotic insufficiency. In this pioneering investigation, physiological tissue parameters were derived from HSI measurements of the proximal site of the anastomosis (esophageal stump) and the distal site of the anastomosis (tip of the gastric conduit) during the thoracic phase of the procedure. Tissue oxygenation (StO_2_), Near Infrared Perfusion Index (NIR-PI), and Tissue Water Index (TWI) showed similar median values at both anastomotic sites. Significant differences were observed only for NIR-PI (median: 76.5 vs. 63.9; *p* = 0.012) at the distal site (gastric conduit) compared to our previous study using an HSI system for open surgery. For all 21 patients, reliable and informative measurements were attainable, confirming the feasibility of HSI-MIS to assess anastomotic viability. Further studies on the added benefit of this new technique aiming to reduce anastomotic insufficiency are warranted.

## 1. Introduction

Carcinoma of the esophagus and the gastroesophageal junction is rising in incidence and prevalence in the Western world [1]. It is difficult to cure because it is asymptomatic in the early stages [2,3]. Often, it is diagnosed at a locally advanced stage, and neoadjuvant (radio-)chemotherapy is required before surgery [1]. Nevertheless, surgical resection of the esophagus with reconstruction by using gastric conduit is still the most effective therapy for locally advanced cancer. Ivor Lewis esophagectomy with gastric pull-up and intrathoracic anastomosis is the most frequently performed surgical procedure in Western Europe [3,4].

Total minimally invasive esophagectomy (TMIE) and robotic-assisted minimally invasive esophagectomy (RAMIE) have been introduced to reduce postoperative complications [5,6]. However, most studies have shown no difference in anastomotic leakage (AL) as compared to open techniques [7,8]. AL is one of the most feared and severe complications after esophagectomy, leading to a significant increase in morbidity and mortality, as well as prolonged hospitalization, resulting in significant utilization of healthcare resources. For the patient, AL means a diminished quality of life, increased cancer recurrence rates, and worsened long-term survival [7,9,10].

An AL is characterized by the Esophagectomy Complications Consensus Group (ECCG) system. It is defined as a “full-thickness gastrointestinal defect involving the esophagus, anastomosis, staple line, or conduit, irrespective of presentation or method of identification” [11]. These leaks are categorized into three levels of severity: Grade I, without any change in therapy; Grade II, necessitating interventional therapy; and Grade III, needing surgical revision [7]. Leaks can be associated with severe infections of the surrounding area of the anastomosis, such as mediastinitis, as well as the emergence of atrial fibrillation, pneumonia, respiratory failure, and the necessity for additional surgery or reintubation [10,12].

The specific pathophysiology and causal factors of AL remain uncertain since it has a multifactorial etiology [13,14]. Besides preoperative malnutrition (albumin < 3.0 g/dL) or underweight patients (BMI < 18.5 kg/m^2^), heart failure, hypertension, diabetes, renal insufficiency, steroids, atherosclerotic calcification of the aorta with local arterial insufficiency [15,16], tobacco use [7,11,17], and neoadjuvant (radio-)chemotherapy [18,19], the poor micro-perfusion of the anastomotic tissue seems to be the leading risk factor for a postoperative AL [9].

Therefore, it is of great significance to identify the site with the optimal perfusion of the tissue while constructing the anastomosis, both proximally at the esophageal remnant stump and distally at the newly formed gastric conduit, to minimize the risk of AL [20]. Imaging techniques, such as Near-Infrared Fluorescence Imaging (NIR-FI) [21,22], performed with an intravenous injection of indocyanine green (ICG) [11,23], or hyperspectral imaging (HSI) for esophageal anastomoses [13], have proven the feasibility of measuring tissue perfusion during surgery [24,25]. However, in addition to the necessary use of a contrast agent, another limitation of NIR-FI is the lack of repeatability of the measurement.

HSI is a non-invasive, contactless method with immense potential that provides intraoperative real-time information about the present tissue within a few seconds [26,27,28,29,30]. The intraoperative region of interest (ROI) is illuminated with light in the visible and near-infrared spectrum, and the light emitted by the tissue is measured. This method gives surgeons more information about the tissue and the various structures they are operating on that the human eye cannot perceive. Currently, it is used to detect the best region for the resection border in gastrointestinal anastomosis and liver resections [31]. However, the field of application has been limited so far, as no minimally invasive system was available for clinical use. The purpose of this study was to investigate whether HSI can be used in minimally invasive oncologic esophagectomies to evaluate tissue perfusion and suitability for esophageal anastomosis to improve patient outcomes.

## 2. Materials and Methods

### 2.1. Patients

In this prospective cohort study, the perfusion of the esophagogastric anastomotic sites of 21 patients (older than 18 years) undergoing either RAMIE procedure (*n* = 10), TMIE procedure (*n* = 8), or hybrid (abdominal part: laparoscopic; thoracic part: open) (*n* = 3) one-time Ivor Lewis esophagectomy due to esophageal and gastroesophageal carcinomas at the University Hospital of Leipzig between 01/2023 and 08/2023 were measured by using the TIVITA^®^ Mini (HSI-MIS, Group 1) hyperspectral intraoperative imaging system, which is described in more detail in Section 2.2. They were compared to measurements using the TIVITA^®^ Tissue (HSI-Open) system in patients (older than 18 years) who underwent RAMIE (*n* = 2), TMIE procedure (*n* = 18), hybrid (*n* = 2), and open (*n* = 8) one-time Ivor Lewis esophagectomy due to esophageal and gastroesophageal carcinomas and esophageal ruptures at the University Hospital of Leipzig between June 2017 and September 2022 (Group 2), covering a different period. All other patients who underwent esophagectomy after ischemic preconditioning of the stomach with a two-staged esophagectomy during this period were excluded. Furthermore, all patients who were unable to consent or were pregnant were excluded from this study. The institutional review board of the University of Leipzig approved this study in ethics agreement 393/16-ek and in an amendment to ethics agreement 026/18-ek. This study was conducted according to the Declaration of Helsinki.

### 2.2. Surgical Procedure and Hyperspectral Imaging

All Ivor Lewis esophagectomies were performed according to standard protocols. The intrathoracic anastomosis (end-to-side) was constructed in double layer technique by using the circular stapler (25 mm and 28 mm in diameter) and oversewing the stapler line using slowly absorbable sutures. In Group 1, both ends of the anastomosis (esophageal remnant stump and gastric conduit tip) were measured by the HSI-MIS. In Group 2, only the gastric conduit tip could be measured by using HSI-Open due to its limited access by the minimally invasive approach. The TIVITA^®^ Mini System by Diaspective Vision GmbH (Am Salzhaff-Pepelow, Germany) is a medically approved HSI system for minimally invasive surgery (HSI-MIS) that is connected with a 30° laparoscopic optic from Karl Storz SE & Co. KG (Tuttlingen, Germany). It is a push-broom scanning hyperspectral imaging system that analyzes the wavelength-specific reflectance of an object in each pixel of the acquired image. The wavelength range from 500 to 995 nm is measured with high spectral resolution in 5 nm steps, leading to one hundred spectral bands [29,32]. This functionality is used to analyze the distinct spectral reflectance properties of different tissues to provide the surgeon with information about tissue oxygenation (StO_2_ in %), which presents the relative oxygenation of the superficial tissue layers. For the deeper tissue layers (4–6 mm), the NIR-PI (from 0–100) provides an estimation of tissue oxygenation, the Organ Hemoglobin Index (OHI from 0–100) indicates the distribution of hemoglobin, and the Tissue Water Index (TWI from 0–100) represents the water content of the observed tissue [28,33,34].

During this pilot study, the HSI-MIS was started and prepared with a sterile plastic covering several minutes before the measurement to minimize the time loss during the surgical procedure. For the measurement of the proximal part of the esophageal remnant stump, the minimally invasive HSI optic was inserted through the assistant trocar during the thoracoscopic phase of the esophagectomy. During the first two operations, the planned anastomotic site was indicated with a laparoscopic instrument while the esophagus was still intact but already mobilized. However, it was subsequently discovered that the use of the bipolar instruments caused excessive superficial charring, impeding adequate HSI measurements and resulting in very low perfusion values, especially for the StO_2_. Subsequent measurements were acquired after esophageal transection and insertion of the circular stapler. Before the measurement, the esophagus and the surrounding area underwent irrigation to minimize artifacts arising from the fresh or coagulated blood and to maximize the exposure of the target structure. All measurements of the esophageal stump were performed in situ with an estimated measurement distance of 5–7 cm (standardized measurement). Furthermore, the regular laparoscopic/thoracoscopic or robotic optic was removed from the situs during data acquisition to minimize any artifacts possibly caused by external illumination. For the measurement of the distal part of the anastomosis (gastric conduit tip), the gastric conduit was pulled out through the mini-thoracotomy, which was done to insert the stapler and construct the anastomosis. Exemplary images of both measurement sites are provided in Figure 1.

To avoid spectral artifacts, the lights of the operating room were turned off. Since taking 1 measurement only takes about five seconds, taking 2–3 measurements of each region was possible without prolonging the procedure. In contrast to our previous in vivo study for the validation of HSI-MIS in a controlled setup [32], no holding arm was used, and the correct distance to the object was subjectively adjusted by the surgeon. A schematic overview of the perfusion imaging systems used and the esophagogastric measurement sites evaluated in vivo by our group is represented in Figure 2 for comparison. Tissue parameters obtained in this study using the HSI-MIS at the distal anastomotic site were compared with measurements obtained using the HSI-Open, which has been described in detail previously [28].

### 2.3. Postoperative Findings and Follow-Up

The follow-up outcome data, including early morbidity and mortality during the first 30 postoperative days, were collected, focusing on ischemic complications of gastric conduit and AL. AL was classified according to the Esophagectomy Complications Consensus Group (ECCG) [7].

Further postoperative diagnostics followed our routine standards. Laboratory parameters were closely monitored, with a special focus on inflammatory parameters.

In the case of any deviations from the expected postoperative course, such as elevated inflammation markers or positive results in the blue dye swallowing test on postoperative day six, further diagnostic assessments, such as gastroscopy and computed tomography scans, were pursued.

### 2.4. Data Analysis

The focus was primarily on the StO_2_, NIR-PI, TWI, and OHI values. These physiological tissue parameters, represented by color-coded images with a finely graduated color scale from red to orange, indicating well-perfused areas, and green to blue, representing poorly perfused areas, were already used intraoperatively by the surgeon to detect the most suitable location for the anastomosis. The color scale ranging from 0 to 100 was previously used in colorectal resections by Jansen-Winkeln et al. [31] and is shown in Figure 3. Following intraoperative measurements of the proximal and distal parts of the anastomosis, the region of interest (ROI) was delineated using the TIVITA^®^ Mini software for analysis, and the data were evaluated postoperatively (Figure 3). Each pixel of a parameter image represents a numerical value of the tissue parameter. For further statistical analysis, the numerical values of each ROI were averaged per patient, and the standard deviation, representing the variability, was calculated. These data were then exported as a text file so that further calculations and visualizations could be carried out with Python. The distributions of these mean ROI values were analyzed in terms of median and quartiles, considering anastomotic site, tissue parameter, and imaging system (HSI-Open vs. HSI-MIS). Additionally, a two-sided Mann–Whitney *U* test was performed for the latter with a significance level of *p* = 0.05. Python 3.6 with SciPy 1.2.3 and Seaborn 0.11.2 library were used for statistical analysis and graphical data visualization, respectively.

## 3. Results

The feasibility of minimally invasive intrathoracic HSI measurements based on the measurement of the proximal and distal parts of the intrathoracic anastomosis following Ivor Lewis esophagectomy (Group 1) is discussed herein, as well as a comparison with the HSI measurements previously conducted using the HSI-Open system (Group 2).

### 3.1. Patients’ Characteristics

The current study includes 21 patients undergoing oncologic esophagectomy (Group 1). All patients were diagnosed with a carcinoma of the esophagus and of the gastroesophageal junction (sixteen with adenocarcinoma, three with squamous cell carcinoma, and one with another carcinoma). The patients had an average age of 60.8 (±7.8) years and an average BMI of 26.4 (±5.5) kg/m^2^. Detailed patient characteristics for both patient groups (HSI-Open vs. HSI-MIS) are presented in Table 1.

### 3.2. Comparison of Intraoperative Perfusion Imaging of the Gastroesophageal Sites (Group 1)

In total, during this study, 58 measurements were taken on the proximal part of the anastomosis (esophageal stump), and 48 measurements were taken on the distal part of the anastomosis (tip of the gastric conduit), with an average of 5.05 (±0.48) measurements per patient. Six measurements of the proximal and twelve measurements of the distal anastomotic component were excluded from the analysis out of the total measurements. They were deemed unusable for evaluation due to motion artifacts (Figure 4), the inability to maintain a measurement distance of 5–7 cm due to anatomic constraints, or field-of-view differences between the HSI measurement and the color image.

One measurement with the HSI-MIS system took 6 to 8 s, and no complications during the measurements were detected. Additionally, it was possible to create the planned anastomosis in the thoracic region with minimum invasiveness in all patients.

The distribution of the mean parameter value inside the ROI of the proximal and distal anastomotic sites is shown in Figure 5. A median (proximal/distal) of 64.7/60.6, 71.2/76.5, 47.9/54.0, and 73.2/48.1 was observed for StO_2_, NIR-PI, TWI, and OHI, respectively. Despite the OHI, the other tissue parameters result in similar median values for both anastomotic sites. Outliers with StO_2_ and NIR-PI values below 30 were observed at the proximal site (esophageal remnant stump) only. However, the ROI mean values of three patients with subsequent anastomotic leakage showed remarkably low StO_2_ and NIR-PI values at the distal site, while the proximal site showed high variability of all tissue parameters. Due to the small number of patients (*n* = 3) suffering from anastomotic leakage in Group 1, no statistical correlation with the measured tissue parameters was performed.

### 3.3. Perfusion Evaluation of the Gastric Conduit by HSI-MIS (Group 1) and HSI-Open (Group 2)

Comparing to the previous measurements with the HSI-Open system and the subsequent evaluation of the HSI-MIS under controlled conditions, it can be observed that performing measurements during thoracoscopic/robotic esophagectomies was feasible in all cases, albeit accompanied by new challenges like motion artifacts or maintaining the measurement distance of approximately 5–7 cm to the tissue.

All of these observations were further corroborated through the comparison of the obtained tissue parameters from the HSI-Open and HSI-MIS systems. Notably, it was observed that, in comparison to the HSI-Open, the HSI-MIS system tends to overestimate the NIR-PI values significantly (*p* = 0.012). In contrast, the StO_2_, TWI, and OHI distributions showed no significant deviation between both systems. The distribution of the ROI mean values at the distal anastomotic site for both HSI systems is shown in Figure 6, and numerical data can be found in Table 2.

### 3.4. Intra- and Postoperative Findings and Follow-Up

AL was identified in nine patients (17%) in both groups. The treatment of AL in Group 1 was successfully conducted using endoscopic vacuum therapy. Additional results can be found in Table 2.

## 4. Discussion

After Thomassen et al. demonstrated that the reliability of HSI data provided by the HSI-MIS is comparable to the HSI-Open system, our study marked the first deployment in a minimally invasive setting during the thoracic phase of esophagectomies [32]. We were able to demonstrate that its use during this procedure was safe and feasible for each patient involved, although the measurement distance was subjectively estimated and not quantified. HSI has been feasible so far for open measurements and has been employed primarily in the creation of gastric conduits. Laparoscopic HSI measurements yielded predominantly good to very good perfusion values, both for the mobilized 2 cm proximal stump of the esophagus and the mobilized and partially vascular-dissected gastric conduit, providing a solid foundation for anastomosis. During our study, we identified only three occurrences of AL in robotic/laparoscopic procedures, yielding an AL rate of 14%. Remarkably, none of these cases required surgical revision. A recent meta-analysis by Casas et al. [22] indicated a marginally reduced AL rate following the application of indocyanine green (ICG) in Ivor Lewis procedures, demonstrating an AL rate of 11%. Additionally, our minimally invasive AL rate of 14% can be contrasted with the 12% AL rate reported in minimally invasive procedures lacking intraoperative imaging, as observed in the multicenter study conducted by Biere et al. [6]. Furthermore, through the comparison with previous HSI measurements of the gastric conduit from June 2017 to September 2022, we demonstrated that in situ measurements during minimally invasive surgeries provide reliable data. Similar to the results under controlled conditions (ex situ and holding device) by Thomassen et al. [32], the HSI-MIS system used in our study appears to significantly overestimate NIR-PI (*p* = 0.012) compared to the HSI-Open.

To the best of our knowledge, this study describes the first in situ measurements using this new type of HSI camera. The demonstrated feasibility marks an important initial step towards clinical integration into regular minimally invasive procedures, aiming to ensure the best possible patient care. Furthermore, the perfusion of the esophageal stump could be objectively evaluated for the first time. Moreover, we detected good perfusion (Median NIR PI > 60) of the esophageal stump, even after mobilizing it by at least 2 cm in all enrolled patients.

Furthermore, intraoperative perfusion assessment was conducted using color-coded HSI images, providing visual representations of perfusion levels to the surgeon rather than relying on precise numerical values available for each pixel. The reliance on color-coded images may present a limitation in detecting subtle variations in perfusion, remaining unclear if this has a significant clinical impact on the occurrence of AL.

Following this, AL is a complex and multifaceted phenomenon influenced by various contributing factors. This study may not comprehensively encompass or account for all potential factors that could contribute to its occurrence. Various elements, such as individual patient health, pre-existing conditions, surgical nuances, and unanticipated events during the recovery phase, may play a role in the development of AL. Attributing its incidence solely to HSI-based perfusion values may be intricate and subject to confounding variables. Limitations of this study include the small number of patients enrolled (*n* = 21), which may limit the generalizability of the findings, and the chosen starting point. Robotic esophagectomies were introduced in our hospital shortly before the study, which understandably resulted in a higher incidence of AL in the first interval of our current analysis (between January and April 2023) compared with the rest of the study period (May to August 2023), with a ratio of 3:0. In this initial period, two of the three involving AL were among the first five robotic Ivor Lewis esophageal resections. Therefore, these cases may not be representative of the relationship between HSI-based perfusion values and the occurrence of AL since it is multifactorial. However, all detected AL were Grade II, indicating interventional therapy.

This pilot study with the HSI-MIS in laparoscopic/thoracoscopic or robotic esophageal resections identified several difficulties during the intrathoracic use of the new device, including motion artifacts resulting from the multi-second recording. The proximity of the device to the heart and lungs during measurements led to motion artifacts in some instances, particularly among patients with low or normal BMI. However, manually holding the device by the surgeon, aided by the stabilization through the trocar, resulted in fewer motion artifacts, contrary to earlier investigations by Thomassen et al. [32] before using a holding device. Nevertheless, these artifacts never reached a degree of severity that would adversely impact the HSI measurements and their subsequent analysis.

The visual feedback of the HSI measurements provided to the surgeon was consistently sufficient for assessing tissue perfusion. However, it cannot be excluded that the motion artifacts and potential overlaps in the measurement results may lead to minor differences, thus limiting the reliability of the measurements.

Furthermore, maintaining a precise 5–7 cm measurement distance was challenging with the minimally invasive approach, as we did not measure it. The distance was estimated based on the clarity and brightness of the real-time image shown while inserting the HSI-MIS. However, each surgeon was capable of estimating the measurement distance based on real-time visual feedback and the associated image brightness, thereby facilitating suitable HSI measurements. Nevertheless, it cannot be ruled out that minor variations in the distance between the device and the tissue being measured may have resulted in slight differences in the HSI data, thereby influencing the study outcome. 

In conclusion, this study demonstrated the applicability of the HSI-MIS system in minimally invasive surgery and provided valuable perfusion data assisting GI surgeons in avoiding severe ischemic complications postoperatively. Furthermore, any mobilization of the esophageal remnant stump did not result in a median NIR PI < 60. Additionally, it has uncovered challenges related to the minimally invasive use of hyperspectral imaging and provides an essential foundation for future multicenter in vivo studies required to validate the results.

## Figures and Tables

**Figure 1 bioengineering-11-00069-f001:**
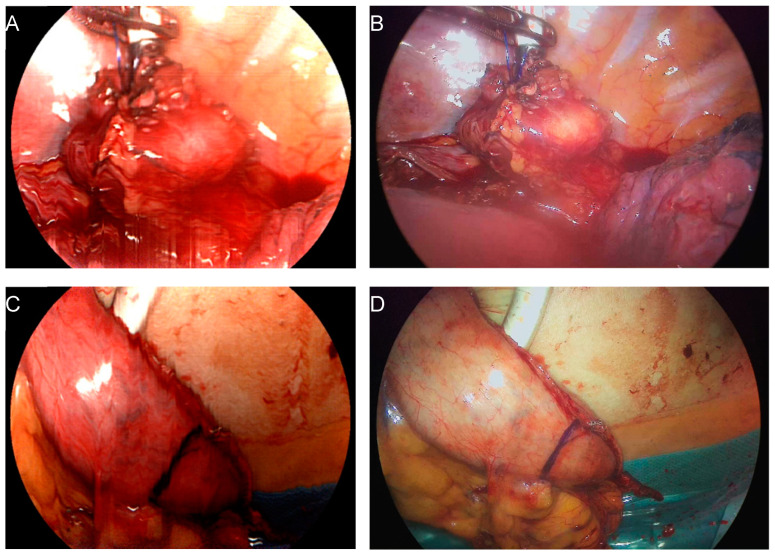
Intraoperative imaging with the HSI MIS system. Reconstructed color image based on acquired HSI data (**A**,**C**). Corresponding color image provided by the RGB sensor of the same system at video rate (**B**,**D**). Imaging of the proximal anastomosis stump after mobilization with inserted stapler head (**A**,**B**) and gastric conduit, the distal end of the anastomosis (**C**,**D**).

**Figure 2 bioengineering-11-00069-f002:**
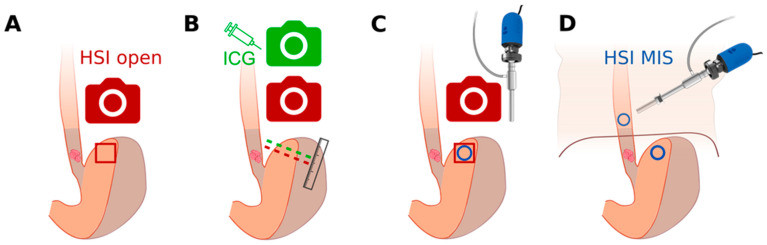
Overview of in vivo studies on HSI of the gastric conduit by our group. (**A**) Evaluation of open surgery HSI for the measurement of ischemic conditioning effects [28]. (**B**) Comparison of open surgery HSI and NIR-FI with ICG for perfusion imaging [24]. (**C**) Validation of the laparoscopic HSI system (HSI MIS) in a controlled setup [32]. (**D**) Feasibility of intrathoracic HSI and evaluation of the obtained tissue parameters regarding anastomotic sites and HSI systems (this work).

**Figure 3 bioengineering-11-00069-f003:**
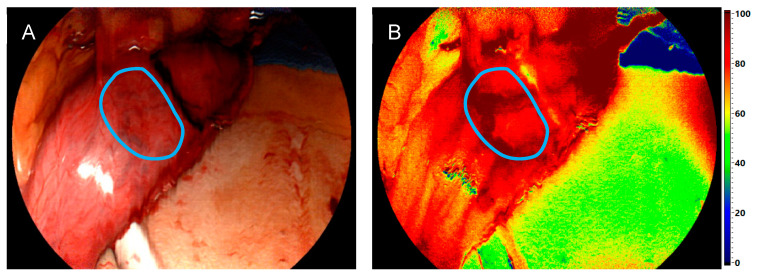
ROI annotation of the distal site of the esophageal anastomosis (gastric conduit). (**A**) Reconstructed color image from HSI data used to define the ROI (blue line). (**B**) Corresponding color map of the NIR PI with ROI for statistical analysis.

**Figure 4 bioengineering-11-00069-f004:**
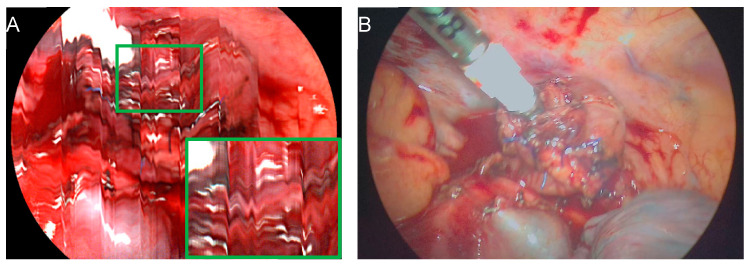
Motion artifacts due to proximity to the heart. (**A**) Cardiac motion during the multi-second acquisition process affects the HSI measurement. (**B**) The corresponding RGB image at video rate is unaffected by cardiac activity.

**Figure 5 bioengineering-11-00069-f005:**
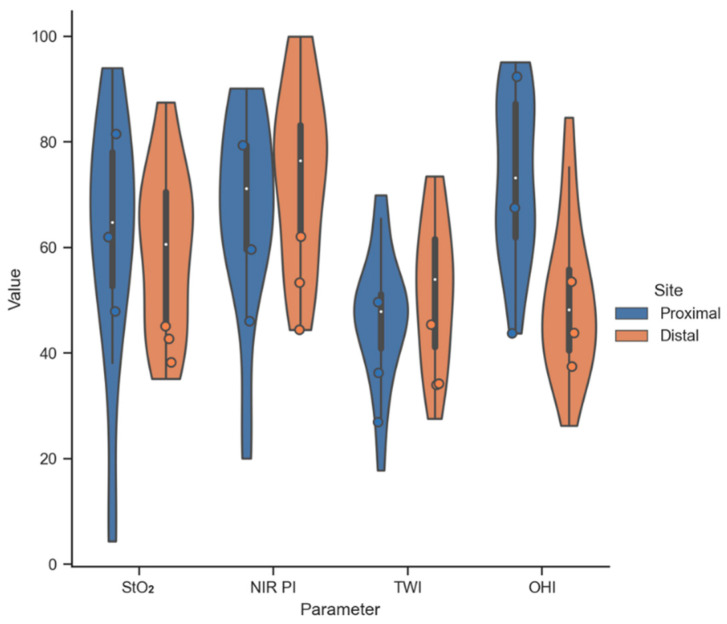
Distribution of the mean parameter value inside the ROI of the gastroesophageal anastomotic site for all patients obtained by HSI-MIS (Group 1). Circles indicate patients with anastomotic leakage (*n* = 3). Distal site: tip of the gastric conduit. Proximal site: esophageal remnant stump.

**Figure 6 bioengineering-11-00069-f006:**
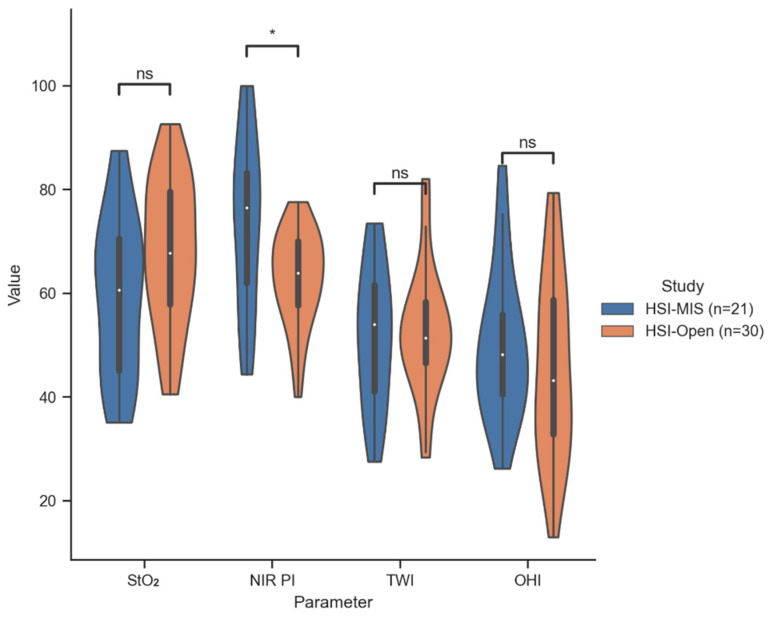
Distribution of the mean parameter value inside the gastric conduit ROI (distal anastomotic site) for all patients measured by HSI-MIS (Group 1) and HSI-Open (Group 2). *: 0.01 < *p* < 0.05; ns: 0.05 < *p* (not significant).

**Table 1 bioengineering-11-00069-t001:** Preoperative findings and demographic data.

	HSI-Open (Jun. 2017–Sep. 2022) (*n* = 30)	HSI-MIS (Jan.–Aug. 2023) (*n* = 21)
**Sex**		
Male/Female	26/4	18/3
**Age (Mean ± SD)**	60.9 (±11.8)	60.8 (±7.6)
**ASA classification**		
ASA I	0	0
ASA II	21	12
ASA III	9	9
**Risk factors**		
BMI (Mean ± SD)	26.5 (±5.3)	27.0 (±5.4)
Diabetes Mellitus	5	7
Hypertension	19	8
COPD	0	2
Smoking	8	5
**Neoadjuvant therapy**		
Chemotherapy	10	12
Radio-chemotherapy	16	7
None	4	0
**Histopathological entity**		
Adenocarcinoma	20	17
Squamous cell carcinoma	7	3
others	3	1

**Table 2 bioengineering-11-00069-t002:** Intra- and postoperative findings and follow-up.

	HSI-Open (Jun. 2017–Sep. 2022) (*n* = 30)	HSI-MIS (Jan.–Aug. 2023) (*n* = 21)
**Type of Operation**		
Robotic	2	10
TMIE ^1^	18	8
Hybrid	2	3
Open	8	0
**Operation duration**		
<360 min	14	18
>360 min	16	3
**UICC Classification**		
0	10	4
I	3	5
II	8	2
III	8	8
IV	1	2
**Size of the circular stapler**		
25 mm	12	9
28 mm	15	12
Suture	3	0
**HSI tissue parameters (Median [Q_1_, Q_2_])**		
StO_2_ (*p* = 0.059)	67.7 [57.8, 79.6]	60.6 [45.1, 70.5]
NIR PI (*p* = 0.012)	63.9 [57.7, 70.1]	76.5 [62.1, 83.2]
TWI (*p* = 0.962)	51.4 [46.6, 58.4]	54.0 [41.1, 61.6]
OHI (*p* = 0.353)	43.2 [32.8, 58.7]	48.2 [40.5, 55.9]
**Anastomotic leak**		
Grade I (%)	1 (3%)	0
Grade II (%)	4 (13%)	3 (14%)
Grade III (%)	1 (3%)	0
**Postoperative pneumonia**	4	0
**Length of stay (Median {range})**	15 {8;68}	13 {9;60}

^1^ TMIE: Total minimally invasive esophagectomy.

## Data Availability

The datasets and codes generated during the current study are available from the corresponding author upon reasonable request.

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
