# Peer review of "Intraoperative Laparoscopic Hyperspectral Imaging during Esophagectomy—A Pilot Study Evaluating Esophagogastric Perfusion at the Anastomotic Sites"

_bioengineering, 2024, doi:10.3390/bioengineering11010069_

Round 1

Reviewer 1 Report

Comments and Suggestions for Authors

The manuscript is well-written and the work is also interesting. The authors should check the typo errors.

Comments on the Quality of English Language

Grammatical and typo errors must be corrected 

Author Response

The manuscript is well-written and the work is also interesting. The authors should check the typo errors.
Answer: Thank you very much for your comment, we checked the manuscript for spelling and grammatical errors and corrected them.

Reviewer 2 Report

Comments and Suggestions for Authors

English and grammar need improvement.

Comments on the Quality of English Language

Dear Author,

Your article "Intraoperative Laparoscopic Hyperspectral Imaging during Esophagectomy – A Pilot Study Evaluating Esophagogastric Perfusion at the Anastomotic Sites" is a novel article which needs a little improvement before publication.

1. English and grammar need improvement.

2. Statistical analysis should be elaborated.

3. Discussion needs referencing which is very poor and that should be correlated with the observations.

4. Limitation of this study should be included.

Thanks

Author Response

Your article "Intraoperative Laparoscopic Hyperspectral Imaging during Esophagectomy – A Pilot Study Evaluating Esophagogastric Perfusion at the Anastomotic Sites" is a novel article that needs a little improvement before publication.
1. English and grammar need improvement.
2. Statistical analysis should be elaborated.
3. Discussion needs referencing which is very poor and that should be correlated with the observations.
4. Limitations of this study should be included.
Answer:

Thank you very much for reviewing our paper and for your notices:

  1. As mentioned above we revised the English and grammar of the manuscript.
  2. We kindly ask the reviewer to specify where a more detailed statistical analysis is desired. Due to the small number of patients, the feasibility of conducting a meaningful statistical analysis was limited.
  3. We added new references and compared our observations with the results of two other studies.
  4. Since we already included some limitations in our discussion, we clarified the existing ones and added more limitations.

Reviewer 3 Report

Comments and Suggestions for Authors

well done and interesting paper to read and to know about

Author Response

well done and interesting paper to read and to know about
Answer:  We would like to thank the reviewer for the positive assessment of our work.

Reviewer 4 Report

Comments and Suggestions for Authors

This is a case study paper evaluating imaging systems. It seems to use the existing software to perform data analysis in Section 2.4. Therefore, it would be better to emphasize (1) the importance of this study and (2) the details of the method used for the data analysis in Section 2.4.
It mentions that there are some limitations, such as motion artifacts resulting from the multi-second recording. It would be good to include these discussions in the paper.
There are a few errors ("reference not found") in the paper; please check the details.

Comments on the Quality of English Language

NA

Author Response

This is a case study paper evaluating imaging systems. It seems to use the existing software to perform data analysis in Section 2.4. Therefore, it would be better to emphasize (1) the importance of this study and (2) the details of the method used for the data analysis in Section 2.4.
It mentions that there are some limitations, such as motion artifacts resulting from the multi-second recording. It would be good to include these discussions in the paper.
There are a few errors ("reference not found") in the paper; please check the details.

We thank you very much for your reviewing our paper and appreciate your comments.
Answer: The description of the data analysis includes all the steps performed. We have supplemented the description and ask the reviewer for specific hints if there are still open points. We had already considered the motion artifacts resulting from the multi-second recording in our discussion, but we have now emphasized this more clearly as a limitation. Furthermore, we checked the manuscript for errors and reported the missing references due to postprocessing to the editor.

Round 2

Reviewer 4 Report

Comments and Suggestions for Authors

The revision is fine.

Comments on the Quality of English Language

NA